# Patient-Specific Guided Osteotomy to Correct a Symptomatic Malunion of the Left Forearm

**DOI:** 10.3390/children8080707

**Published:** 2021-08-17

**Authors:** Femke F. Schröder, Feike de Graaff, Anne J. H. Vochteloo

**Affiliations:** 1Orthopedisch Centrum Oost Nederland, Centre for Orthopaedic Surgery, 7555 DL Hengelo, The Netherlands; f.schroder@ocon.nl (F.F.S.); a.vochteloo@ocon.nl (A.J.H.V.); 2Techmed Centre, Faculty of Science and Technology, University of Twente, 7522 NB Enschede, The Netherlands

**Keywords:** forearm malunion, osteotomy, patient-specific guides, 3D

## Abstract

We present a case report of a 12-year old female with a midshaft forearm fracture. Initial conservative treatment with a cast failed, resulting in a malunion. The malunion resulted in functional impairment for which surgery was indicated. A corrective osteotomy was planned using 3D analyses of the preoperative CT-scan. Subsequently, patient-specific guides were printed and used during the procedure to precisely correct the malunion. Three months after surgery, the radiographs showed full consolidation and the patient was pain-free with full range of motion and comparable strength in both forearms. The current case report shows that a corrective osteotomy with patient-specific guides based on preoperative 3D analyses can help surgeons to plan and precisely correct complex malunions resulting in improved functional outcomes.

## 1. Introduction

Forearm fractures are common among children and are often treated with closed methods (i.e., closed reduction and cast immobilization). Although bones in young children are generally forgiving, they can heal in an abnormal position. i.e., a malunion. Malunions occur in about 15% of fractures, and can lead to, among other complaints, pain, carpal and distal radioulnar joint instability, reduced range of motion, and in the long term, osteoarthritis [1,2,3]. Corrective osteotomy, a surgical method to restore normal bone anatomy, should be considered if the malunion results in functional impairment [4]. 

Historically, corrective osteotomies were planned based on radiographs. These images can be sufficient in the case of simple fractures, as angular and translational deformities can generally be assessed in 2D. However, malunited forearm fractures are often complex, involving deformities in multiple anatomical planes. As found in a study of Miyake et al. [5], complex malunited forearm fractures seem to have rotational deformities in a range from 115 degrees of pronation to 15 degrees of supination. Plain 2D radiographs generally do not provide adequate information about rotational deformities and are therefore not suitable for the preoperative assessment of the malunion and planning of correction osteotomies [6,7]. 

In recent years, there has been an increased interest in the use of 3D analysis and printing in (paediatric) orthopaedics [7,9,10]. Nowadays, a corrective osteotomy is often preceded by preoperative planning based on 3D analyses of the malunion. The introduction of 3D preoperative planning and printing of patient-specific guides has significantly improved the interpretation of complex fractures. Several studies have shown that the correction of malunited forearm fractures can be precisely planned and performed using 3D analyses and printing of guides, which subsequently resulted in improved functional outcomes [11,12,13,14,15].

We present a case report of a 12-year old female with a malunited midshaft forearm fracture treated with patient-specific guided corrective osteotomies of radius and ulna. The case is a perfect example of how preoperative planning and printing of patient-specific guides create the possibility to restore the preoperative anatomical position and function of complex malunited fractures.

## 2. Case Description

### 2.1. Patient Case and History

In June 2020, a 12-year old girl presented at the emergency department after falling on her left forearm while jumping on a trampoline. After physical examination, radiographs of the left forearm were obtained (Figure 1A). The patient was diagnosed with a mid-shaft antebrachial fracture without dislocation which was treated conservatively with a cast. During follow-up, the fracture consolidated, however, with a secondary displacement of the fracture. Conservative treatment was prolonged. 

Three months after the initial fracture, the arm was still painful and function of the forearm was limited (elbow flexion/extension 145-0-0, pronation/supination 10-0-10, palmar flexion/dorsal flexion 60-0-60). A video of the preoperative range of motion is added in the Appendix A. This complicated daily activities and sports. Radiographs (Figure 1B) showed bowing and dorsal angulation of both radius and ulna. The patient was diagnosed with a symptomatic malunion of the left forearm. Therefore, 3D preoperative analyses were performed and a corrective osteotomy using patient-specific sawing and drilling guides was scheduled: this process takes about a month. At the request of her parents, the operation was scheduled three months after the analyses. Pain and function of the forearm had not improved compared to three months post-fracture.

### 2.2. Preoperative Planning & Guide Design

First, a CT scan (slice thickness of 0.6 mm) of both forearms was made. Based on the CT data, 3D models (Figure 2A,B) of the left and right forearm were created using Mimics (Materialise, Leuven, Belgium) software. To obtain a 3D model, automatic threshold-based bone segmentation was performed. Afterwards, the segmentation of the bones was checked, made solid, and labelled. 

The uninjured contralateral forearm (right side) was mirrored over the injured forearm (left side) to use as a model of the anatomical desired position of the left forearm (Figure 2A,B). As a result, the rotational deviation of the injured side with respect to the uninjured side could be measured. Based on the centre of gravity of the radius, there was an increase of 15° of dorsal inclination, 11° of pronation, and 2° of radial deviation. For the ulna, the differences were smaller, with 2° of dorsal inclination, 8° of pronation, and 6° of ulnar deviation Then, a double osteotomy for the radial part and a single osteotomy of the ulnar part was performed (Figure 2C,D) and the distal parts of the radius and ulna were rotated and translated into the desired anatomic position (Figure 2C,D). The 3D preoperative analyses were discussed and authorised by a multidisciplinary team (orthopaedic surgeons and a technical physician).

Patient-specific guides were designed based on the preoperative 3D plan in 3-Matic 13.0 (Materialise, Leuven, Belgium) software. First, drilling guides were designed (Figure 3A). These guides direct the drilling of holes to eventually fixate the plate in the right position after the osteotomy. Second, the sawing guides were created. These guides are used to perform the osteotomies (Figure 3B). Additionally, both guides contained 3 holes for 1.4 mm K-wires for fixation of the guides during the procedure. Once the guides were created, the 3D models of the preoperative radius and ulna, and the postoperative desired radius and ulna, were made print ready. The 3D printed models of the pre- and postoperative radius and ulna were used during the procedure to assure the right position of the guides and fixation and to provide insight into the procedure. 

Finally, the designed patient-specific guides were exported as a stereolithographic (STL) file and sent to a 3D printing company which 3D laser-printed the designed patient-specific guides from medically certified polyamide powder. The patient-specific guides were post-processed and packaged for hospital sterilisation and sterilized in house according to our standard clinical guidelines. 

### 2.3. Surgical Procedure

The patient was operated under general anaesthesia. The left arm was positioned on an arm table and a tourniquet was applied. First, the radius was corrected through a volar Henry [16]. The patient-specific drilling guide was positioned on the radius. The position of the guide was confirmed using the surface anatomy of the bone and a 3D printed model of the radius. Once the position of the guide was verified, 1.4 mm K-wires were used to fixate the drilling guide. Subsequently, the screwholes were made in the radius through the guide. Subsequently, the drilling guide was replaced by the sawing guide using the same K-wires, and again the correct position of the guide was verified. The osteotomy was performed with an oscillating saw. The sawing guide and K-wires were removed and the radius was corrected to the planned position using a plate and screws.

The standard direct approach to the ulna was used, and similar to the correction of the radius, the patient-specific drill guide was used first, followed by the osteotomy guide. Subsequently, both guides were removed. 

For both the radius and ulna, plates (2.4 mm straight LCP plates, Synthes) were placed on the pre-drilled holes and fixated with cortical and locking screws. The correction of the radius and ulna were verified under fluoroscopy. The range of motion was passively tested (elbow flexion/extension 145-0-0, pronation/supination 85-0-75, palmar flexion/dorsal flexion 80-0-80). The patient received an above elbow cast for 2 weeks. 

### 2.4. Postoperative Results

The radial and ulnar corrections were evaluated using the CT-scan based 3D preoperative planning and postoperative radiographs of the forearm. After surgery, a CT scan was not obtained to avoid unnecessary radiation. To allow for image-based comparison, the preoperative 3D models were evaluated in the same view as the postoperative radiographs (Figure 4).

Clinically, the patient started visiting an experienced hand therapist (S.v.B.) every two weeks, starting at two weeks after surgery. Six weeks after surgery, the range of motion of the left forearm was comparable with the contralateral side (elbow flexion/extension 150-0-5, pronation/supination 90-0-90, palmar flexion/dorsal flexion 80-0-90). A video of the postoperative range of motion is added to the Appendix A. Three months after surgery, the patient was pain-free with full range of motion, and comparable grip strength of both hands. The radiographs showed full consolidation (Figure 1C). 

## 3. Discussion and Conclusions

This case report presents the results of a corrective osteotomy of the forearm using 3D preoperative analysis and patient-specific guides in a patient with a malunited midshaft forearm fracture. The 3D preoperative analysis gives insight into the rotational and translational malalignment: without the planning and the drilling and sawing guides, it is almost impossible to perform a sufficient correction. The patient-specific guides ensure that the surgery will be performed according to plan, resulting in a correction in three planes of the malunion. 

A corrective osteotomy may be considered when functional impairment persists as a result of a malunited fracture. In this specific case, the arm was painful, and the function of the forearm was limited three months after trauma. Although there has been debate about (the timing of) corrective osteotomies of malunions of forearm fractures in children, previous studies have shown favorable results. A review by Roth et al. (Roth et al. [17]), based on individual patient data of 11 cohort studies including 71 participants, revealed that corrective osteotomies provided a mean gain in forearm rotation of 77° (68° to 86°). In our case, we found an even better result, with full recovery of forearm rotation from 10° to 90°. Moreover, the study of Roth et al. revealed that both a younger age at osteotomy (median age 11 years at trauma) and a shorter time until osteotomy (median of 12 months between trauma and osteotomy) were associated with a better functional outcome. In our case, the patient was 12 years of age, with only 7 months between trauma and surgery. The relatively young age and short time between trauma and surgery may have contributed to the good functional outcome in our case study. 

As mentioned earlier, malunited forearm fractures are often complex, involving deformities in multiple anatomical planes. The malunion presented here also included deviations in all anatomical planes for both the radius and ulna. These complex fractures are generally difficult to assess and plan based on plain 2D radiographs. Several studies have used 3D techniques to precisely plan and perform correction osteotomies of malunited forearm fractures. Byrne et al. [12] investigated the use of 3D planning and patient-specific guides in five consecutive patients with a diaphyseal forearm fracture and found an increase in both supination and pronation and a significant improvement in pain and grip strength. Miyake et al. [7] included 20 patients with a forearm malunion and found an improvement in angular deformity and an improvement in the mean arc of forearm motion. Kataoka et al. [14] investigated the use of computer-planning for the correction of malunited diaphyseal forearm fractures in four patients and found an improvement in angular deformity on X-ray, range of forearm rotation, and grip strength. These studies demonstrate that the use of 3D planning and printing of patient-specific guides helps surgeons to precisely perform corrective osteotomies of complex malunited fractures. This is supported by the findings of a review by Roth et al. [17], who showed that the use of 3D computer-assisted techniques is a predictor of superior functional outcome after corrective osteotomies of forearm malunions. 

We are aware of the possible limitations of the current study in that it only describes the results of one patient. The results presented here, however, provide a complete picture of this specific case. Furthermore, one might consider the disadvantages of the use of 3D preoperative planning and patient-specific guides, as these techniques are cost and time-consuming. Another limitation is the necessity of a preoperative CT scan for 3D planning and printing. The additional dosage of the required CT scan compared to plain radiographs might be considered a problem, especially in children. Because of this, we have not obtained a postoperative CT scan to evaluate the performed corrective osteotomy. On the other hand, however, it has been shown that the use of 3D planning and printing reduces operating time, intraoperative blood loss, and fluoroscopic exposure [9].

The current case report shows that performing a corrective osteotomy with patient-specific guides based on preoperative 3D analyses for specific malunited fractures in children is of added value and makes it possible to precisely correct complex malunions of the forearm and closely mimic the pre-injured situation. The use of these computer-assisted techniques can help surgeons to plan and accurately perform corrective osteotomies resulting in improved functional outcomes. 

## Figures and Tables

**Figure 1 children-08-00707-f001:**
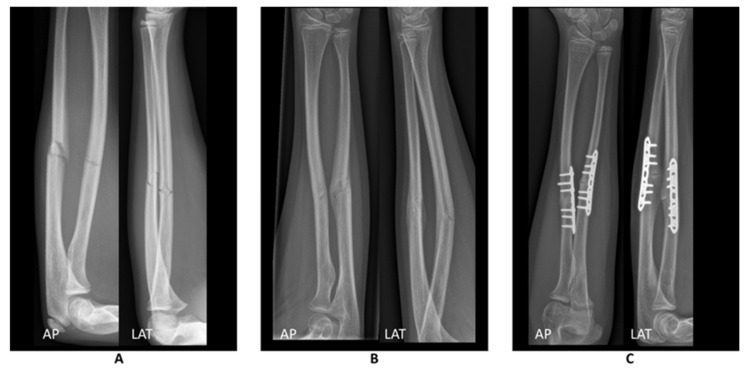
Radiographs of the forearm in anterior posterior (AP) and lateral (LAT) directions (**A**) at initial trauma, (**B**) 3 months after initial trauma where the fracture is consolidated and the arm shows bowing and dorsal angulation, and (**C**) after patient-specific guided correction.

**Figure 2 children-08-00707-f002:**
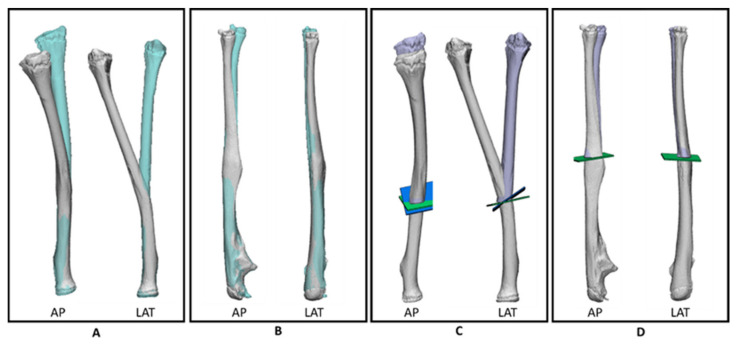
3D models based on the preoperative CT scan in anterior posterior (AP) and lateral (LAT) position of (**A**) the injured left radius (grey) and the mirrored contralateral radius (blue), (**B**) the injured left ulna (grey) and the mirrored contralateral ulna (blue), (**C**) planned double osteotomy of the radius where the green and blue planes represent the first and second sawing planes, respectively. After correction, the injured radius (grey) can be repositioned towards the desired (purple) position, and (**D**) planned single osteotomy of the ulna with the green plane representing the sawing plane.

**Figure 3 children-08-00707-f003:**
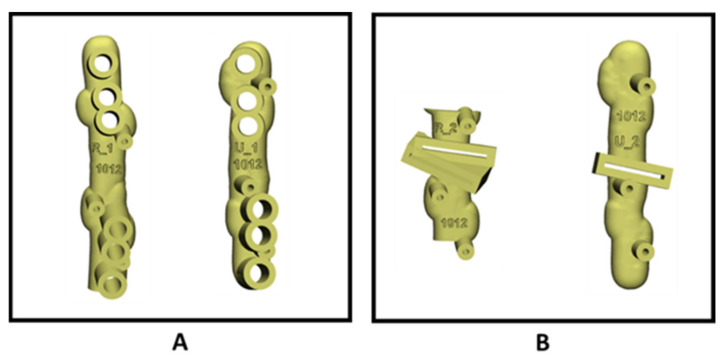
(**A**) Patient-specific drilling guides for the radius (left) and ulna (right), and (**B**) patient-specific saw guides for the radius (left) and ulna (right).

**Figure 4 children-08-00707-f004:**
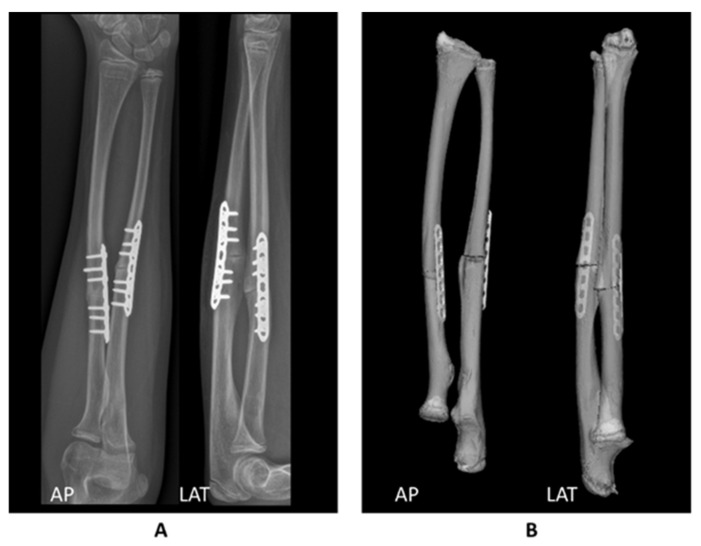
Comparison between (**A**) the postoperative radiographs and (**B**) the 3D planned postoperative position. Based on visual comparison, the correction was performed as planned.

## Data Availability

Since this is a case report all relevant data is presented in the result section.

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
