# Peer review of "Patient-Specific Guided Osteotomy to Correct a Symptomatic Malunion of the Left Forearm"

_children, 2021, doi:10.3390/children8080707_

Round 1

Reviewer 1 Report

This is a well-balanced case report. I think a few corrections may make this article more readable.
1) The patient visited the author’s clinic after three months of the initial trauma, but the surgery was performed 7 months after the trauma. The authors should clarify whether the four-month delay came from the 3D planning procedure or not.
2) Please clarify how the cost and time of this 3D planning procedure.
3) Line 110, ulna  radius
4) Line 111, the standard ulnar approach may be better. 
5) Line 114, VA locking plate of 2.4mm of Synthes may be the distal radius plate. Please correct.
6) A forearm corrective osteotomy may need a bone graft for the potential defect at the osteotomy site. Did the case use a bone graft (local or from elsewhere) or both forearm bones were fixed tightly without any bone graft? If a bone graft was not used, it may be another advantage of this 3D planning procedure.

Author Response

This is a well-balanced case report. I think a few corrections may make this article more readable.

  1. The patient visited the author’s clinic after three months of the initial trauma, but the surgery was performed 7 months after the trauma. The authors should clarify whether the four-month delay came from the 3D planning procedure or not.

Thank you for your comments.  Indeed, the analyses were performed 3 months post fracture, and the surgery was scheduled 7 months post fracture. The main reason for the interval between analyses and surgery was the wish of the patient and her parents: they wanted to wait another 3 months after the analyses. The 3D planning process and printing of the guides takes about 4 weeks.

To clarify this, we changed the text in manuscript (page 2; lines 60-63):

“Therefore, 3D preoperative analyses were performed and a corrective osteotomy using patient-specific sawing and drilling guides was scheduled: this process takes about a month. On request of her parents, the operation was scheduled three months after the analyses. Pain and function of the forearm had not improved compared to three months post fracture.”

 2. Please clarify how the cost and time of this 3D planning procedure.

Thank you for your question. The 3D planning of surgeries is done in house and takes about 10 hours of our technician: this costs about €225. Printing of the guides is done outside our clinic at a company to fulfil all European regulations and CE marking. This part of the process takes about 2 weeks and costs approximately €150. Thus, in total the costs are around €400.

Therefore, time from CT to printed guides in our set-up is 3-4 weeks. We mentioned the time in this new version (see point 1)

We have not mentioned costs in the paper as this is very different from patient to patient, and hospital to hospital, even within in our country. Internationally, the differences are even larger, due to different regulations and different financial structures of health care systems. 

  1. Line 110, ulna à radius

The text has been changed as suggested.

  1. Line 111, the standard ulnar approach may be better. 

Thank you for your suggestion. The text was changed to (page 4; lines 114-115): The standard direct approach to the ulna was used, and similar to the correction of the radius, the patient-specific drill guide was used first, followed by the osteotomy guide.”

  1. Line 114, VA locking plate of 2.4mm of Synthes may be the distal radius plate. Please correct.

We did use straight 2.4 LCP plates for both radius and ulna. To clarify this we changed the text to (page 4; lines 117-118):

“For both the radius and ulna, plates (2.4 mm straight LCP plates, Synthes)  were placed on the pre-drilled holes and fixated with cortical and locking screws.”

  1. A forearm corrective osteotomy may need a bone graft for the potential defect at the osteotomy site. Did the case use a bone graft (local or from elsewhere) or both forearm bones were fixed tightly without any bone graft? If a bone graft was not used, it may be another advantage of this 3D planning procedure.

Thank you for your question. For these procedures, we routinely do not use bone grafts as we always aim to keep bone contact and secure the osteotomy with a solid osteosynthesis.  

The planning of the osteotomies are performed in such a manner that bone contact is secured.

Reviewer 2 Report

There is an important point that isn’t so clear in this work and that influences the acceptance for the credibility of the paper:

  • 56: 3 months after the surgery…arm still painful and function limited…61: corrective osteotomy was scheduled
  • Figure 1…63: 3 months after initial trauma
  • 150….7 months between trauma and surgery…

…all that means: 3 months after trauma there was a painful arm and bowong and dorsal angulation on X Rays and for these reasons an osteotomy was scheduled. The osteotomy was performed 4 months later.

The questions are:

  • The arm was painful also after 7 months from trauma?
  • There was not any remodeling of the bone in this period?
  • Weren’t prescribes physical therapys during this time (magnetotherpy for example)?
  • Can we see x- ray examination or CT scan after 7 month from trauma?

Author Response

Reviewer 2

There is an important point that isn’t so clear in this work and that influences the acceptance for the credibility of the paper:

56: 3 months after the surgery…arm still painful and function limited…61: corrective osteotomy was scheduled. Figure 1…63: 3 months after initial trauma. 150….7 months between trauma and surgery…all that means: 3 months after trauma there was a painful arm and bowing and dorsal angulation on X Rays and for these reasons an osteotomy was scheduled. The osteotomy was performed 4 months later.

The questions are:

  1. The arm was painful also after 7 months from trauma?

Thank you for your comment. As we also mentioned in the comments of reviewer 1:

She had pain and limited function three months after the fracture. Therefore the 3D analyses were performed and the operation was suggested. However, the patient and her parents  wanted to postpone the operation for personal reasons. At time of surgery (7 months post fracture) she still had pain and the same, limited function of her forearm.

To clarify this, we changed the text in manuscript (page 2; lines 60-63):

“Therefore, 3D preoperative analyses were performed and a corrective osteotomy using patient-specific sawing and drilling guides was scheduled: this process takes about a month. On request of her parents, the operation was scheduled three months after the analyses. Pain and function of the forearm had not improved compared to three months post fracture.”

There was not any remodeling of the bone in this period?

Thank you for your question. Three months after trauma, there was full consolidation and a clear malunion (see Figure 1 (B)). As no further X rays were made in the interval between 3 and 7 months, remodeling of the bone could not be assessed. However, the drilling and sawing guides, that were based on the CT scan that was made 3 months post fracture fitted perfectly during surgery. From this we conclude that no remodeling has happened, otherwise these guides would not have fitted perfectly.

Weren’t prescribes physical therapys during this time (magnetotherpy for example)?

The patient was not prescribed physical therapy, as of the large rotational limitation we did not think this was useful. Furthermore, patient and parents did not want any additional conservative treatment.  

  1. Can we see x- ray examination or CT scan after 7 month from trauma?

As was addressed on your question about remodelling of the bone: we do not have additional imaging such as an X-ray or CT-scan between 3 and 7 months post trauma as the imaging at 3 months after trauma showed consolidation, and the patient did not show any signs of clinical changes. Therefore we did not want to expose her to additional radiation.

Round 2

Reviewer 1 Report

All the concerns were answered appropriately.

Reviewer 2 Report

If for the other reviewer and for the Editor it's not a problem if we cann't see an imaging 7 months after the fratture but online that one after 3 in a 12 years old child growing up and with a big potential of remodeling and healing, for me the xorrecrions are ok